# Maps and metrics of insecticide-treated net access, use, and nets-per-capita in Africa from 2000-2020

Amelia Bertozzi-Villa [1,2,3 ✉], Caitlin A. Bever[2], Hannah Koenker[4], Daniel J. Weiss [1,5], Camilo Vargas-Ruiz[1], Anita K. Nandi [3], Harry S. Gibson [3], Joseph Harris[1], Katherine E. Battle [1,2,3], Susan F. Rumisha[1,6], Suzanne Keddie[1], Punam Amratia[1], Rohan Arambepola[3], Ewan Cameron[1,5], Elisabeth G. Chestnutt[3], Emma L. Collins[3], Justin Millar[3], Swapnil Mishra [7], Jennifer Rozier[1], Tasmin Symons[1], Katherine A. Twohig[3], T. Deirdre Hollingsworth [3], Peter W. Gething [1,5,9] & Samir Bhatt[7,8,9]

Insecticide-treated nets (ITNs) are one of the most widespread and impactful malaria interventions in Africa, yet a spatially-resolved time series of ITN coverage has never been published. Using data from multiple sources, we generate high-resolution maps of ITN access, use, and nets-per-capita annually from 2000 to 2020 across the 40 highest-burden African countries. Our findings support several existing hypotheses: that use is high among those with access, that nets are discarded more quickly than official policy presumes, and that effectively distributing nets grows more difficult as coverage increases. The primary driving factors behind these findings are most likely strong cultural and social messaging around the importance of net use, low physical net durability, and a mixture of inherent commodity distribution challenges and less-than-optimal net allocation policies, respectively. These results can inform both policy decisions and downstream malaria analyses.

[1] Malaria Atlas Project, Telethon Kids Institute, Perth Children's Hospital, Nedlands, WA, Australia. [2] Institute for Disease Modeling, Seattle, WA, USA. [3] Big Data Institute, Li Ka Shing Centre for Health Information and Discovery, University of Oxford, Oxford, UK. [4] Tropical Health, Baltimore, MD, USA. [5] Curtin University, Bentley, WA, Australia. [6] National Institute for Medical Research, Dar es Salaam, Tanzania. [7] MRC Centre for Global Infectious Disease Analysis, Department of Infectious Disease Epidemiology, Imperial College London, London, UK. [8] Section of Epidemiology, Department of Public Health, University of Copenhagen, Copenhagen, Denmark. [9] These authors jointly supervised the work: Peter W. Gething, Samir Bhatt. ✉email: abertozzivilla@idmod.org

Insecticide-treated nets (ITNs) are one of the most widespread, impactful, and cost-effective tools for combating malaria transmission in Africa, averting an estimated 450 million cases from 2000 to 2015[1–3]. Mass ITN distributions have been a core component of malaria control for almost two decades, over which period billions of dollars of aid have flowed into ITN production and distribution, resulting in the delivery of more than two billion nets to households across the continent. While the broad importance of ITNs is generally uncontested, many questions remain regarding optimal net allocation strategies, net owners' decisions on when to use and discard nets, and how to position ITN campaigns within a complex landscape of risk, cost, and impact. While others have addressed these questions locally and specifically, this analysis aggregates data and fills gaps across space and time to provide context and detect patterns in ITN distribution, utilization, and retention for 40 malaria-endemic countries in Africa from 2000 to 2020.

In summarizing the coverage of ITN distributions, two indicators are generally used: access, which measures what proportion of the population could sleep under a net assuming two people per net[4]; and use, which measures what proportion of the population does sleep under a net. These two metrics are often combined to calculate the use rate, defined as use among those with access. A final indicator, nets-per-capita (NPC), tracks net volume among the population and is primarily used for procurement purposes[5,6]. Throughout this paper, we will refer to metrics specifically by name, or use the term "coverage" to indicate any combination of metrics. All results in this analysis are reported among the population at risk of malaria, defined here as anyone living in a zone with stable *Plasmodium falciparum* malaria transmission[7].

Initially, ITN interventions were targeted at children under five and pregnant women[8]. However, targets for ITN coverage have since expanded to include the entire population at risk[9], though "universal coverage" targets are commonly set to 80% for both access and use[5,10]. In recent years, language around coverage targets has softened to acknowledge the heterogeneous intervention needs of different settings, though universal coverage remains the official policy[11].

Together, these coverage metrics encapsulate three key challenges to optimal ITN performance: distribution, utilization, and retention. Distributing nets from manufacturers to homes across Africa presents a massive supply-chain and commodity challenge that hinders sufficient net acquisition for many households. At higher coverage levels, there is some evidence of ITNs being overallocated to more accessible households that already have sufficient nets, leaving coverage gaps for less protected households[4,12]. While ITN utilization is often high among those with access[13–16], heterogeneities have been observed across space[14], seasons[16], age groups[17,18], and genders[19]. There is some evidence of a small proportion of nets being misused for fishing[20–22] and older nets being repurposed[23], but these activities do not occur at a sufficient scale to cause concern from a malaria control perspective. Finally, many studies have suggested that median net retention times may be dramatically shorter than the 3-year duration presumed by mass campaign schedules[24–27], though others have estimated longer survival times[28–30]. Understanding the history and spatial distribution of ITN access, use, and NPC is crucial for effective ITN policy planning.

This analysis utilizes a Bayesian mixed modeling framework built upon data from net manufacturers, national programs, and cross-sectional household surveys over the past 20 years to estimate the history of ITN coverage metrics in 40 sub-Saharan African countries. This approach includes two main steps (Fig. 1). First, a national-level "stock-and-flow" mechanistic model tracks the distribution, acquisition, and loss of ITNs by triangulating data from the three sources listed above. This step estimates both ITN retention times and ITN crop, the total number of nets in the community. Second, a series of geospatial models disaggregates this national time trend down to 5-by-5-km pixel resolution by estimating ITN access deviation, the local variation from national mean access, and use gap, the local difference between access and use. From access and use, the ITN use rate or use among those with access can be determined. NPC is calculated similar to access via a NPC deviation model.

Previous publications have reported on ITN coverage metrics over time[12,31] or space[1,14,32,33]. This analysis presents high-resolution maps of ITN access, use, and NPC from 2000 to 2020. Our findings support the evidence that ITN use is high among those with access, but that insufficient access continues to hinder progress toward coverage targets. We show that universal ITN access is difficult to achieve due to the volume of nets needed and is subsequently difficult to maintain due to shorter-than-expected median net retention times. As an additional challenge, the relationship between NPC and access shows saturation effects, which make high levels of access extremely costly to achieve. Resolving these allocative inefficiencies could dramatically increase ITN access in countries with high per-capita net coverage. In addition to reporting access, use, and use rate independently, we report the results of a relative gain analysis to quantify the improvements in ITN use that could be achieved by improving access versus improving the use rate. These maps and national estimates may be useful for benchmarking and comparison exercises, and are publicly available for use as covariates in other malaria-related analyses.

## Results

**Access and use**. Aggregating across all countries in this analysis, after over a decade of steady increases, all ITN coverage metrics have plateaued since 2016 (Fig. 2, bottom left, all rates are reported among the population at risk of malaria). In 2018 and 2019, continent-level coverage metrics declined for the first time since mass ITN distributions began, with net crop decreasing from a peak of 360 million (95% confidence interval 345–376) in 2017 to 337 million (317–357) in 2019 and access decreasing from 56.3% (54.1–58.8) to 51.0% (48.5–53.6) over the same time period. Mass campaigns in 2020 are estimated to have set a net crop record of 364 million (341–383), but access has continued to plateau at 51.8% (48.8–54.8). The use rate has increased with small fluctuations across the time series examined, from a low of 71.3% (67.6–75.1) in 2004 to a 2020 estimate of 87.1% (83.1–90.3).

Nationally, while most countries have made extensive progress in ITN access and use since 2000, coverage levels remain below the WHO targets. Figure 2 shows estimates of ITN access and use at a monthly time scale across all countries in this analysis. The dotted horizontal line represents 80% access or use to indicate the current normative WHO benchmark for universal coverage. Fourteen of 40 countries have ever achieved this coverage, and in 2020, Benin, Mali, Niger, Togo, and Uganda were the only countries estimated to achieve over 80% use (though in some cases lower confidence bounds were below 80%, and Niger has no survey data to support its estimates). Burundi, Gambia, and Ghana have seen particularly steep coverage declines in recent years, while Mali, Mozambique, Togo, and Uganda have sustained higher coverage. Use aligns closely with access in almost all countries. Madagascar, which has a strong culture of net use despite limited access, shows slightly higher use than access in many years, suggesting the widespread practice of more than two people sharing a net. Ghana, Nigeria, and Zimbabwe experienced a considerable use gap beginning in 2010, but only Ghana has maintained that gap to the present day, with a use rate

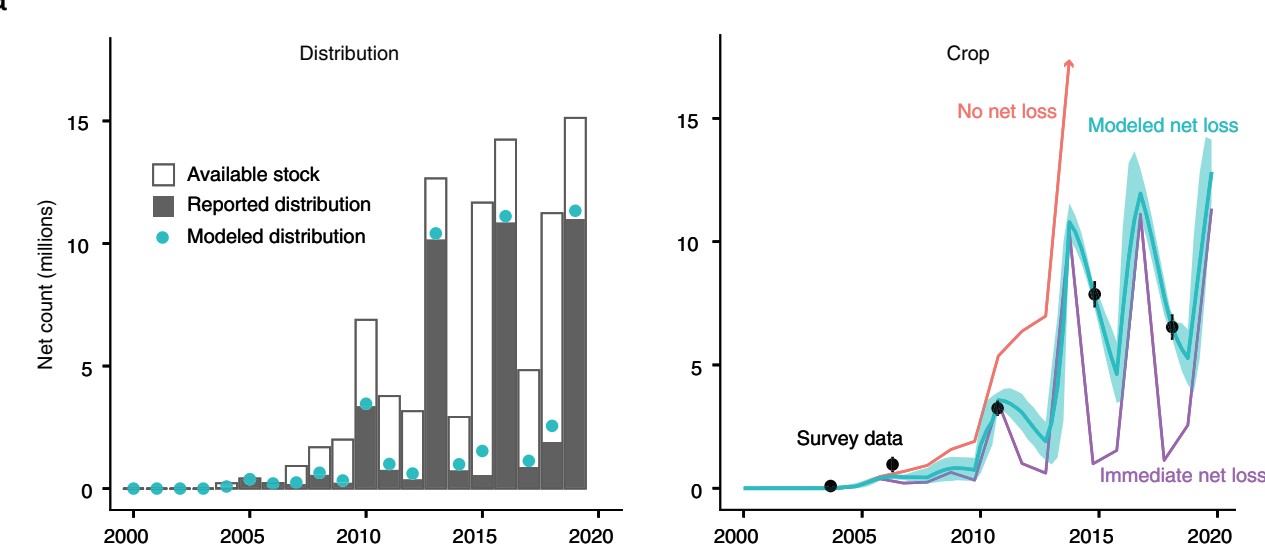

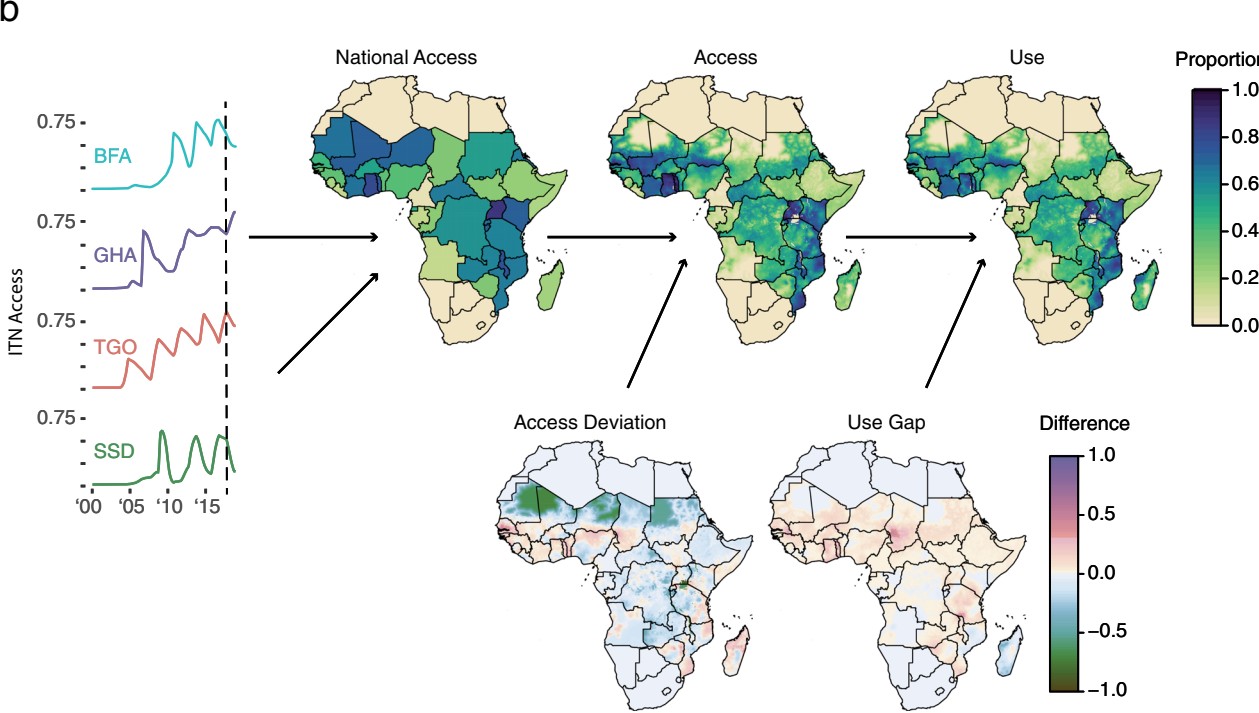

**Fig. 1 Insecticide-treated net (ITN) model summaries. a** Mechanistic "stock-and-flow" model. For each country (Burkina Faso shown for reference), the number of nets distributed must be no less than the reported distribution count and no more than the available stock (solid bars, left). Net loss follows an S-shaped curve whose steepness is fitted according to survey data (right). If nets were never discarded, net crop would increase cumulatively with every distribution (red line), whereas if nets were discarded immediately, net crop would equal net distribution (purple line). The fitted curve (blue line and 95% confidence interval) balances these two extremes. **b** Geostatistical regression model. After net crop time series are converted to net access time series, geospatial regression models are run on the difference metrics of "access deviation" and "use gap." Final maps of ITN access are calculated by adding national access and access deviation, while final maps of ITN use are calculated by adding access and use gap. Maps of nets-per-capita are calculated similarly to access. BFA Burkina Faso, GHA Ghana, TGO Togo, SSD South Sudan. These four access time series are shown as examples, but 40 countries are included in the analysis.

of 61.8% in the most recent 2019 survey. This corresponds to a use gap of 25.2 percentage points. The modeled estimate of use gap in Ghana for 2019 was 22.8 (19.5, 25.8) percentage points, while the aggregated all-country estimates for use rate and use gap were 87.1% (83.3, 90.1) and 5.69 percentage points (4.53, 7.02), respectively.

A number of studies[16,17,34,35] have established that the ITN use rate often tracks seasonal changes in climate and malaria transmission. While the present analysis was not designed to explore these subannual trends, our results show stronger seasonality in use than access even though both models include the same covariates. This finding supports the importance of

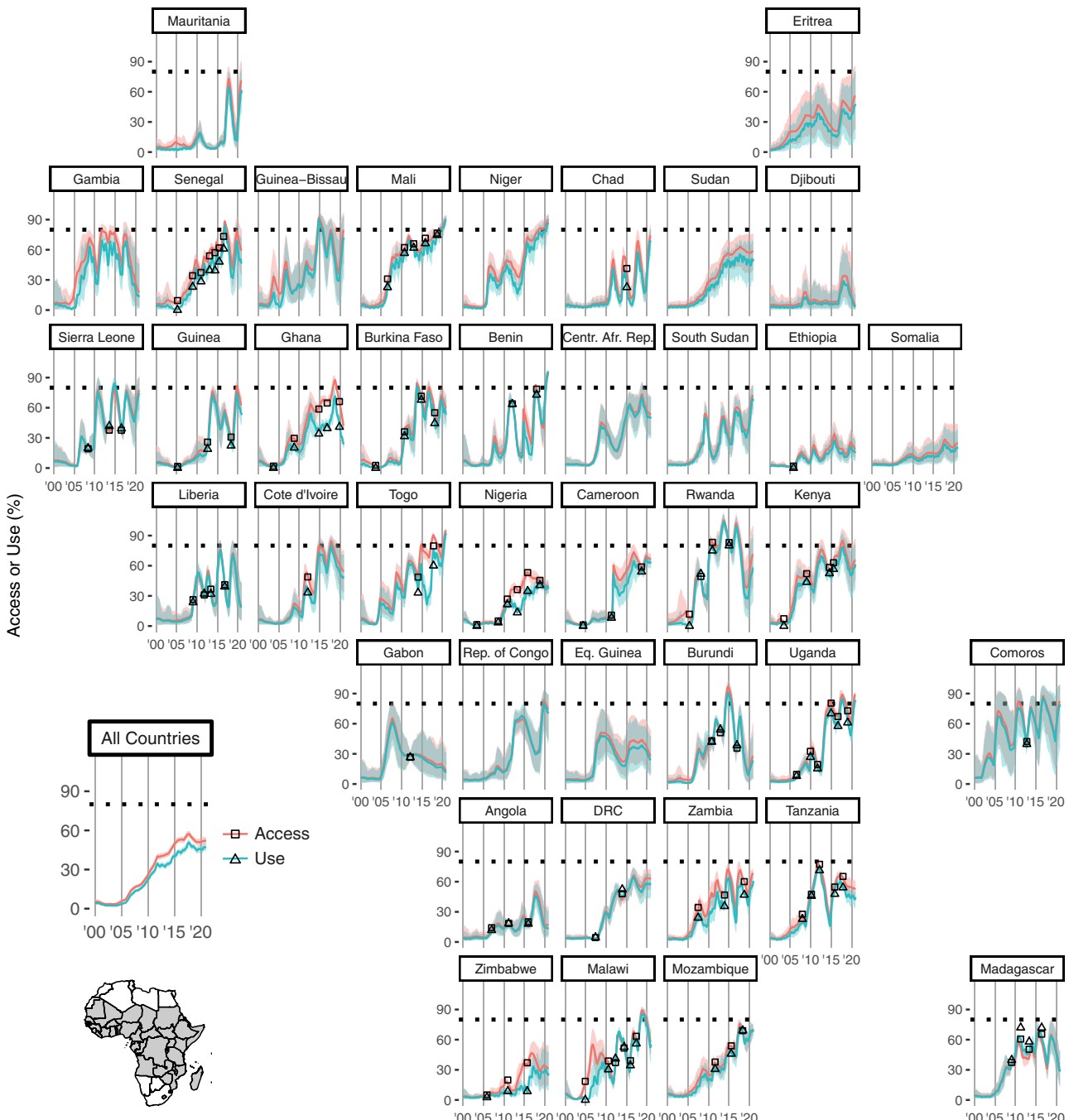

**Fig. 2 National-level insecticide-treated net access and use time series.** National-level mean estimates of access (red curves) and use (blue curves) among the population at risk are shown at a monthly time scale. The dotted lime indicates 80%, commonly used as a benchmark for universal coverage. Squares and triangles indicate nationally representative survey values of access and use, respectively. Only surveys that include spatial information are shown here; surveys included in the stock-and-flow model but not the geospatial model can be found in Supplementary Table 1.l. Shaded areas represent 95% conrfidence intervals. Spikes in coverage indicate mass net distribution campaigns, with coverage subsequently declining due to attrition. The seasonality of use is evident in the curvature of the blue lines. Centr. Afr. Rep. Central African Republic, Rep. of Congo Republic of Congo, DRC Democratic Republic of the Congo.

considering season, climate, and perceived risk when interpreting net use data.

Figure 3 highlights subnational heterogeneity in every ITN coverage metric in 2020. Some of this variation is clearly associated with risk-based distribution strategies, such as the stark north-to-south gradients of access and NPC in Mali, Chad, and Sudan. Other subnational heterogeneity may instead suggest areas experiencing challenges to distribution campaigns, such as

much of central and northeast Nigeria. Some countries are more homogeneous in their access and NPC estimates, most notably Angola, Zimbabwe, Ethiopia, Somalia, Gabon, Benin, and Togo. Estimated spatial patterns of the ITN use rate differ substantially from those of access and NPC, with higher values overall and notably heterogeneous patterns in Ghana. Western Tanzania stands out as an area of concern, but this may be an artifact of data collection. For both the 2015–16 Demographic and Health

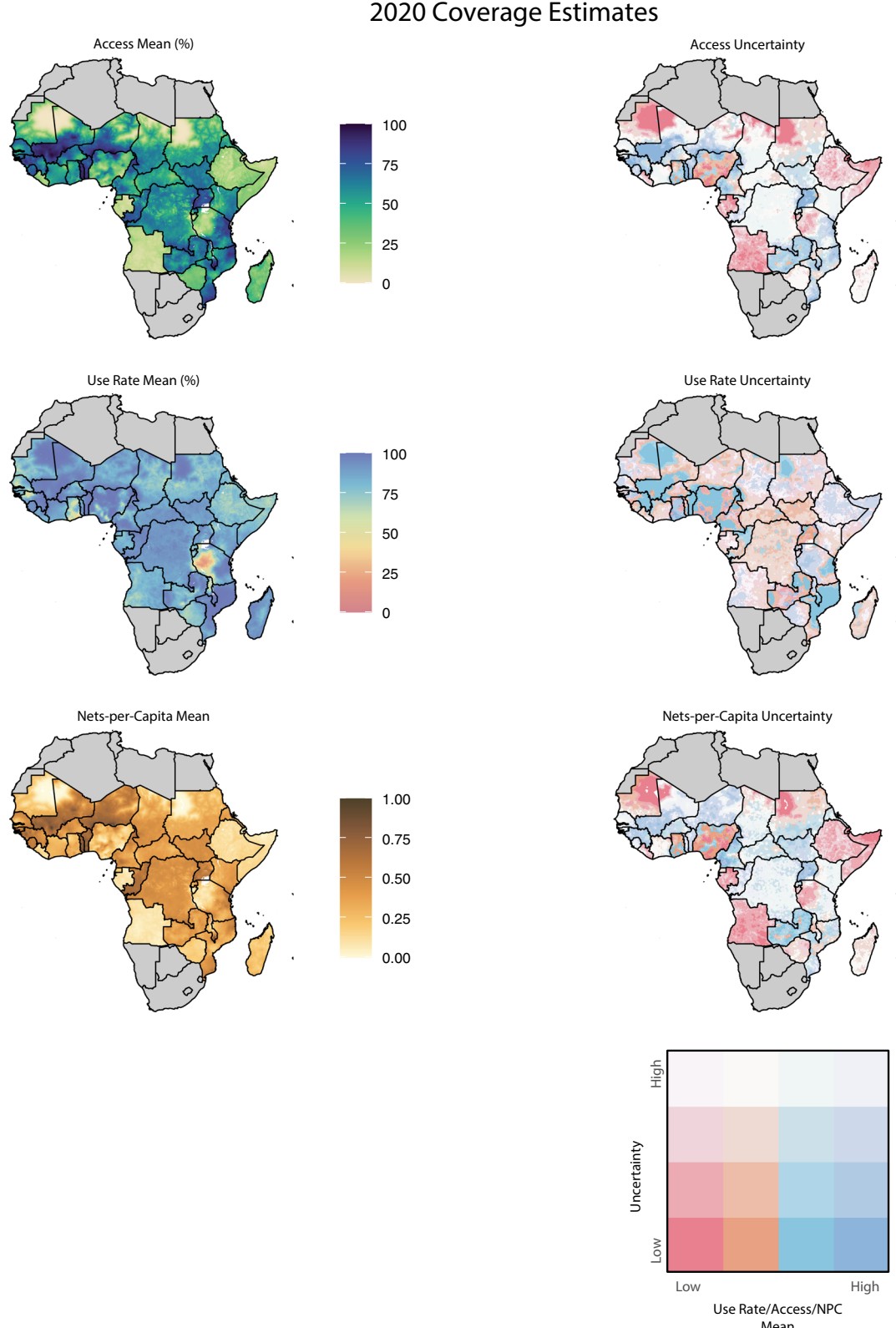

**Fig. 3 Maps of insecticide-treated net access, use rate, and nets-per-capita (NPC) in 2020, with associated uncertainty.** Access is defined as the proportion of the population that could sleep under a net, assuming one net per two people. The use rate is defined as use among those with access. The uncertainty maps use a bivariate scale to convey information about both mean estimates and magnitude of uncertainty. Quantiles of 95% confidence interval width are represented by saturation level, with the most uncertain values having the lowest saturation. Quantiles of mean values are represented by hue, with low values in pink and high values in dark blue. For example, access estimates are most uncertain in the Democratic Republic of the Congo and Zimbabwe in 2020, but there is considerable certainty that much of northern Mozambique is in the third quartile of use rate.

## Access-NPC Relationship, 2020

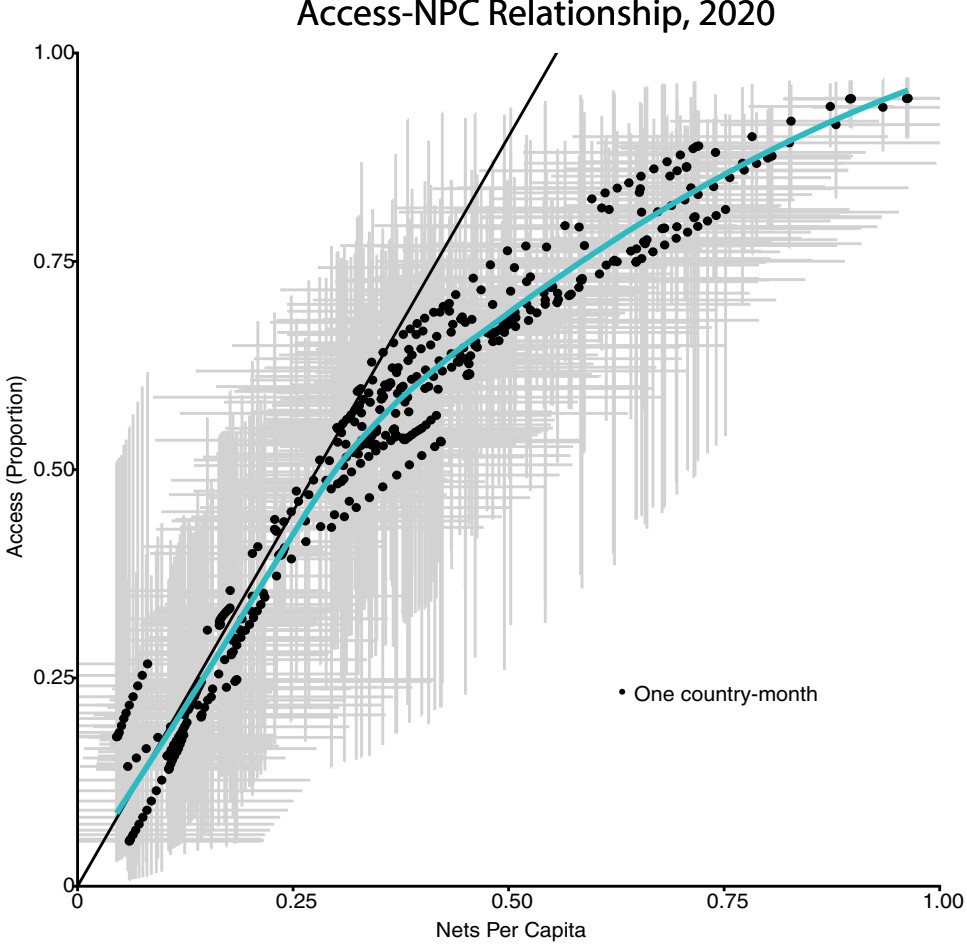

**Fig. 4 Relationship between access and nets-per-capita (NPC) in 2020.** Each dot represents a modeled country-month, with black dots indicating mean values and gray bars indicating 95% confidence intervals for each metric. The solid line has a slope of 1.8, showing the relationship between NPC and access presumed by population-based net procurement decisions. The blue curve shows a Loess fit through the estimated points. The expected linear relationship between NPC and access holds well at low coverage levels, but the true relationship tapers off at NPC values greater than 0.25 and access over 0.5. This plateauing of access despite high numbers of nets distributed per capita suggests inefficiencies or redundancies in net distribution at these coverage levels, such that those who should be receiving nets are still left without access.

Survey and 2017 Malaria Indicator Survey, nets were distributed in western Tanzania over the same time period as survey data collection. Surveys conducted just before large net distributions might explain some of the effect observed in this region.

The uncertainty plots in Fig. 3 present population-binned mean values and 95% confidence interval width for each metric. More saturated colors represent narrower uncertainty, while the pink-to-blue spectrum indicates mean values. For example, in Saharan regions such as Mauritania and northern Mali we are confident that net access was in the lowest quartile in 2020, while in eastern Tanzania we are moderately certain that it was in the highest quartile. In the Democratic Republic of the Congo, Zimbabwe, and Kenya, which have not conducted household surveys in several years, estimates are highly uncertain. Full time-series maps of all metrics and associated uncertainty are available at the Malaria Atlas Project website (https://malariaatlas.org/research-project/metrics-of-insecticide-treated-nets-distribution). This website also hosts an interactive visualization of results and uncertainty, which features an option to plot uncertainty in terms of exceedance probabilities (Supplementary Fig. 2.8) as a complement to the relative uncertainty shown here.

**Access versus NPC.** NPC and access are highly correlated metrics, but their relationship features important nonlinearities

that pose challenges for net allocation (Fig. 4). When planning mass campaigns, WHO recommends that countries procure one net per 1.8 people at risk (0.56 NPC) to ensure universal access while accounting for households with an odd number of residents[6,36]. The tacit assumption is that ITN access will scale linearly with the number of nets distributed. Our analysis demonstrates that this assumption holds at low coverage levels. However, at NPC levels above ~0.25, the relationship between NPC and access begins to plateau, such that attaining coverage levels of 0.5 NPC corresponds to ITN access well below 0.75, instead of the predicted value of 0.9. This pattern suggests substantial misallocation of nets at higher coverage levels, such that some households are left with insufficient protection despite the total number of nets in a country continuing to increase (Supplementary Fig. 2.7). The results shown here represent modeled estimates, but this relationship is also evident in the survey data alone (Supplementary Fig. 2.6).

**Net retention time.** The model generates separate national estimates of median retention time for long-lasting insecticide-treated nets (LLINs) and conventional ITNs (cITNs). Because LLINs have been the primary net distributed in Africa for many years, we focus on LLIN retention here. Our models estimate shorter LLIN lifespans than the 3-year retention time assumed on an

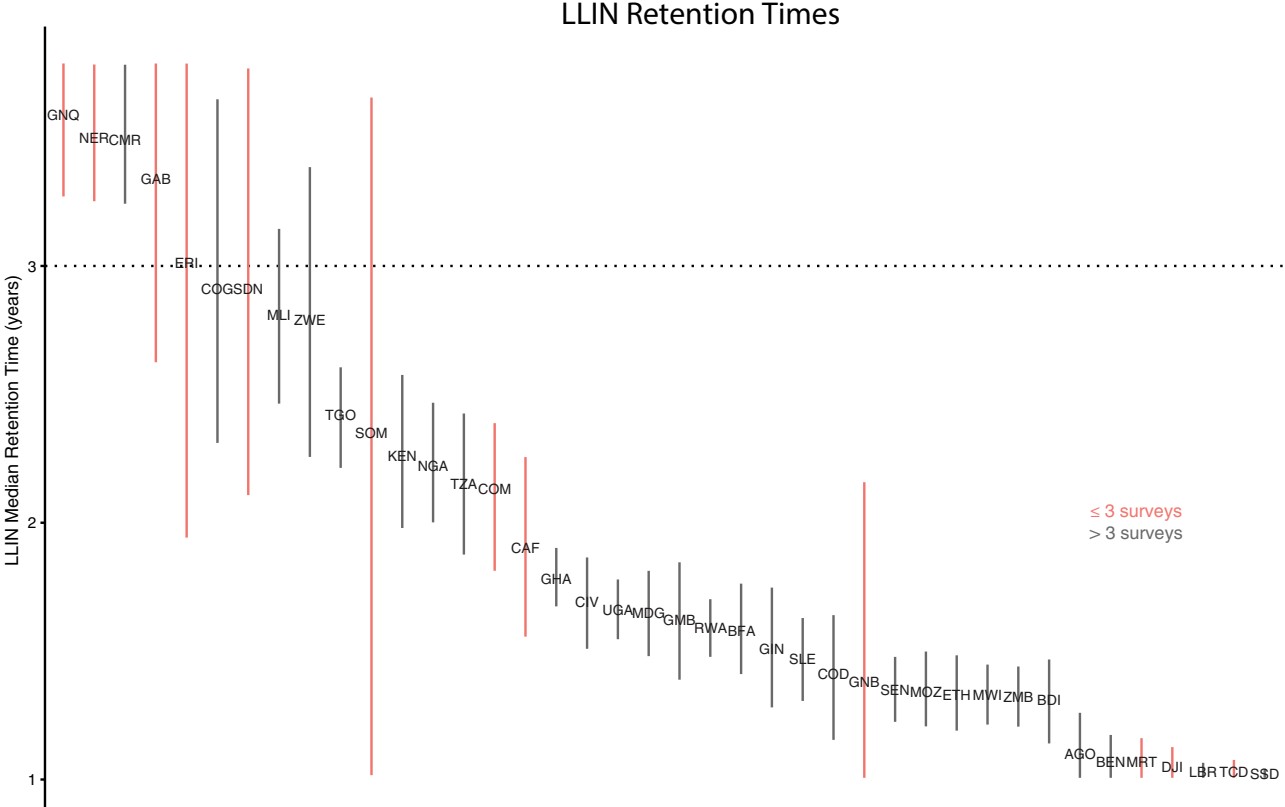

**Fig. 5 Long-lasting insecticide-treated net (LLIN) median retention times.** Stock-and-flow estimates of median LLIN retention time by country, ordered from highest to lowest . Countries are labeled by ISO3 code. Country labels are positioned at mean parameter values, while vertical bars indicate 95% confidence interval width. Countries with fewer surveys have less stable model fits (see Supplementary Section 1.6); those having fewer than three surveys are indicated in red. The lower bound of LLIN retention time was capped at 1 year during model fitting. Supplementary Table 1.7 shows all data for this figure in numerical format and maps ISO3 codes to country names.

average by WHO and other policy-making bodies (Fig. 5 and Supplementary Table 1.7)[37]. Of the 40 countries in this analysis, 35 show LLIN median retention times of under 3 years, with an overall median value of 1.64 years (IQR 1.33–2.37). Many of the countries with the most extreme retention times are also those with the fewest surveys available for fitting, and thus these values may not be reliable (see sensitivity analysis in Supplementary Note 1.6). Among countries with stable model fits, Cameroon is the only country with a median retention time over 3 years (3.49 years, 95% CI 3.24–3.78). Thirteen countries have median retention times whose mean and upper confidence bound are below 1.5 years, from Mozambique (1.34 years, 1.21–1.50) to South Sudan (1.02 years, 1.01–1.04). The model prior on retention time was bounded at 1 year, suggesting that some countries might show even shorter median retention times in an unbounded setting. However, given the small number of surveys in these countries, such results might indicate underspecified models rather than truly short retention times. See Supplementary Note 3.3 for caterpillar plots of the retention time parameter for all countries.

**Relative gain: optimizing access or use?** The relative gain analysis supports other evidence that lack of access is the primary barrier to universal coverage. The top row of Fig. 6 shows the estimated use map for 2020. The second row shows what the use map would look like if access was held constant and the use rate was 100% (left), versus if the use rate was held constant but access was 100% (right). The bottom row represents these two plots in terms of the percentage point increase in use that could be

achieved by increasing the use rate as opposed to increasing access.

In 2020, with the exception of western Tanzania and scattered areas of the Sahel, lack of access to a net was a far greater barrier to coverage than deficiencies in use. While the overall impact of increasing access is greater than the impact of increasing use rate for all years, the spatial patterning of this relative gain varies substantially from year to year. For example, in 2015, it is not Tanzania but Zimbabwe and Chad where use rate is a larger barrier than access[14] (Supplementary Fig. 2.12). The season in which surveys are conducted may play a substantial role in the volatility of use rate estimates.

**Discussion**

The goal of this analysis was to fully characterize ITN access, use, and NPC over space and time for 40 sub-Saharan African countries. We found that ITN distributions have increased enormously since the year 2000 and those who own nets tend to use them, but a combination of insufficient net volume, distribution inefficiency, and short retention times keep ITN access and use below the WHO targets of 80% coverage.

Despite prodigious effort on the part of national malaria programs, mass net distributions have rarely been able to distribute sufficient nets to all those in need to attain universal coverage. Historically large campaigns completed in 2020 despite COVID-related disruptions pushed several countries over the 80% threshold for access and use. These include Benin, Mali, Niger, Togo, and Uganda. If current trends continue, some countries, such as Sierra Leone, Burkina Faso, Kenya, Malawi, and

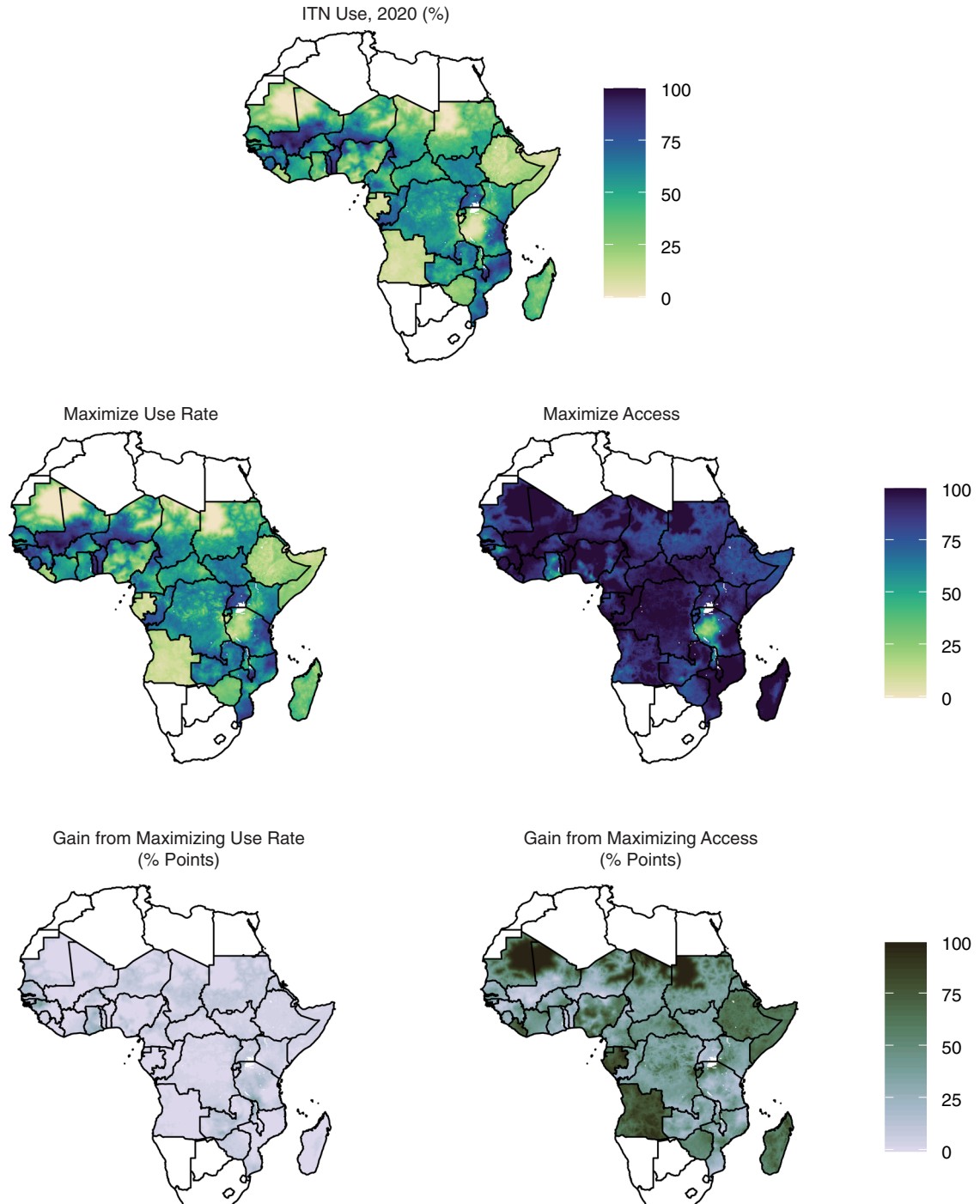

**Fig. 6 Magnitude of change in insecticide-treated net (ITN) use possible from increasing use rate versus increasing access.** The top row shows estimated ITN use in 2020. The second row shows what use could be if access remained unchanged and the use rate were set to 100% (left), compared to if the use rate remained unchanged and access was set to 100% (right). The final row shows the magnitude gain in use from each of these two scenarios. With few exceptions, increasing access has a larger impact than increasing the use rate.

Mozambique, may distribute sufficient nets for universal coverage within one or two mass campaigns, whereas other countries such as Cote d'Ivoire, Liberia, and Tanzania have lost ground or stalled progress. However, official policy is now shifting away from mandates of universal ITN coverage and toward recommendations for more holistic and locally tailored strategies that utilize a range of tools to provide universal malaria protection[11,38]. This would reduce the relevance of the 80% target in favor of a more nuanced national and subnational approach.

Our analysis adds to the evidence[4,12] of net misallocation at higher coverage levels, leading to lower-than-expected ITN access for a given number of NPC. Some amount of misallocation is likely unavoidable as a corollary to well-established econometric relationships that show dramatically increasing intervention costs at high coverage levels[39], but considerations of how to mitigate this effect are still worthwhile. One known source of net misallocation is the mass campaign practice of capping the number of nets any single household can obtain, disadvantaging large

households[5], but logistical, geographic, or cultural factors may also play a role. Such misallocation may have an immense negative effect on ITN impact and cost-efficacy as a malaria intervention.

While some studies have observed median net lifespans longer than 3 years[28,29,40,41], the bulk of existing evidence supports the notion that median net retention is commonly lower than 3 years[24–27,30,42]. The primary motivation for discarding a net in these studies was the perception that it was too torn, with even a modest amount of net damage often regarded as unseemly or untidy[43]. The ITN literature upon which the 3-year timeline is founded focuses on nets' anti-mosquito properties, not attrition[44]. This raises a crucial distinction between the technical lifespan of nets based upon their ability to repel and kill mosquitoes despite tearing and other damage, and the effective lifespan of nets, which in many locations seems to be shorter due to lack of physical integrity (Smith et al., in preparation). Policy solutions to this discrepancy could include more frequent net distributions, shifting to the production and distribution of more physically durable nets[45–50], or using community engagement to change cultural notions about the acceptability of owning and using a torn net. The optimal combination of these strategies will depend heavily on local context and culture.

We find little evidence of low use among those who own ITNs, especially in areas of high malaria transmission, implying that in most settings there are strong social and cultural norms around the importance of net use when possible. This is consistent with a large body of literature on ITN use rates[13–16]. In this literature, lower use rates are commonly reported in times and geographies of lower actual and perceived risk, such as during the dry season or in highland areas. Use rates in these studies are also highest among children and women, and lowest among teenagers. Our analysis supports these seasonal and geographic trends, but does not disaggregate by sex or age. Data from a recent study of net durability across five years and seven countries support our findings that use of available nets remains steady even as net retention declines[51], though the modeling accompanying this analysis suggests a large benefit to increasing the use rate.

Both our estimates and national survey data show surprisingly low use rates in Nigeria in the early 2010s and in Ghana from 2005 to the present. The 2013 Nigeria DHS report suggests that this large use gap may have been due to conducting the survey in a season of low malaria transmission, though an independent study also found low rates of net use in Nigeria in the mid-2010s[13]. The 2019 Ghana MIS report cites three primary reasons for lack of net use: that the net was an "extra" or being saved for later, that it was too hot to sleep under a net, or that other anti-mosquito methods were preferred. In both countries, use rates are lowest in coastal and urban areas, suggesting correlations with increased wealth, improved housing, lower perceived risk, and access to malaria treatment and alternative malaria prevention strategies.

The combined mechanistic and geospatial modeling framework presented here was necessary to accurately capture spatio-temporal trends in the data, but also lends itself well to the exploration of different distribution scenarios. This feature proved valuable for policy planning in the face of the COVID-19 pandemic, in which it was necessary to consider counterfactual scenarios of reduced malaria intervention coverage[52]. With this framework, we were able to mechanistically assign unique scenarios to each country to understand how different campaign strategies would affect overall coverage.

This analysis has several limitations. As a continent-level analysis conducted by and for stakeholders based in Europe and

North America, this work perpetuates both the "foreign pose" and the "foreign gaze" of global health[53] and should not override the expertise of local researchers and policymakers. Causation is complicated in the relationship between net coverage and malaria—while increased net access and use are known to reduce malaria burden, net distributions are also often targeted to areas where burden is highest. This analysis estimates ITN coverage without incorporating information about malaria transmission in order to allow these estimates to be used as predictors for burden estimation[52,54]. The coverage metrics reported here do not capture the waning of net efficacy over time due to insecticide decay and physical deterioration, meaning that estimates of ITN use are not fully capturing ITN effectiveness. In the stock-and-flow model, we assume that all nets reported as distributed by NMCPs are successfully provided to households. If, in reality, the net distribution pipeline is leaky and a fraction of nets are not successfully provided, net crop will be overestimated in some years, leading to an underestimate of net retention times. This analysis does not capture differences in net brand or private sector acquisition and distribution of nets[27].

Large data gaps in both space and time impact these results in a number of ways. Conducting the nationally representative surveys upon which this analysis relies is a Herculean effort, requiring many thousands of work hours from thousands of people. The cost of such an effort prohibits many countries from conducting these surveys more than once every few years. As such, each year of this analysis includes geospatial information for only a handful of countries and each country has geospatial information for a median of 4 years. In addition to increasing the overall uncertainty of our results, data sparsity may bias the results in a number of ways. First, our geospatial model assumes that the outcome variables (access deviation, NPC deviation, and use gap) vary smoothly over space, allowing neighboring countries to inform each other's estimates. While this smoothness assumption has some face validity (Supplementary Fig. 2.1), there is no mechanistic reason why this relationship must hold. Second, several countries only have one or two nationally representative surveys capturing ITN access, and we are unlikely to accurately estimate the ITN time series or LLIN retention times in these cases (see sensitivity analysis in Supplementary Note 1.6). Third, in countries with multiple surveys, data collection commonly occurs in the same season every survey. Since net use is often seasonal, estimates of use rate may be consistently biased toward the use rate of a single season in these countries. Fourth, in this model framework surveys conducted just before large net distributions would lead to predictions of incorrectly low access and use rates, as may have occurred in western Tanzania from 2016 onward. Conducting additional surveys in future years, especially in countries with sparse data at present, would dramatically improve estimates. While net distribution data are far more complete than survey data, it is only provided at the annual and national scale for most countries. Since mass distribution campaigns are typically conducted over several months on a region-by-region schedule, this analysis could gain much precision by incorporating finer-scale space and time data around distributions.

When transforming these estimates into policy decisions, a local understanding of at-risk populations is crucial for effective net distribution. This continental-level analysis defines population at risk simplistically as the total population in any pixel with the potential to sustain malaria transmission, when in fact the relative risk of any individual varies based on urbanicity, wealth, housing type, use of other interventions, and local transmission pathways, to name just a few. In addition, population estimates themselves are often uncertain and may impact our results. For

example, the 2020 mass net campaign in Benin found a 13.5% discrepancy between expected and true population[55].

In future iterations of this model framework, we hope to incorporate waning net efficacy, take a more nuanced view of populations-at-risk, and incorporate data (where available) on subnational and subannual net distributions. As policy recommendations shift from a focus on universal coverage with ITNs to universal coverage with an appropriate mix of interventions[11], we plan to use this infrastructure in combination with other mechanistic modeling to advise on optimal intervention packages at national and local scales.

In combination with locally collected data and the strategic expertise of campaign organizers, we hope that these estimates can provide useful context for country programs planning ITN distributions. The evidence provided here of short net retention times implies a potential immense benefit to procuring more durable nets as they reach the market in coming years[45–48,50], and may suggest the utility of informational campaigns encouraging people to retain slightly torn nets. Countries that are successfully distributing large numbers of nets but experiencing limited gains in access might consider strategy changes to ensure that nets are reaching their intended users. Researchers who may want to adapt this framework to their context can find all code and documentation on GitHub (https://github.com/bertozzivill/map-itn-cube/tree/publication-2021), and we welcome collaboration on such projects.

In malaria policy, understanding the past is a crucial component of planning for the future. We present maps of ITN access, use, and NPC from 2000 to 2020 at 5-by-5-km spatial resolution. We show that ITN distribution campaigns have contributed enormously to coverage in the last two decades, and that the large majority of people who own nets use them. We also show, however, that current net distribution volume is insufficient to maintain high coverage in a landscape of low retention rates and inefficient distribution modalities. This flexible and robust modeling framework for net coverage estimation is appropriate for both long-term planning and emergency strategizing in times of crisis, and we hope that this tool can continue to play a valuable role in informing ITN policy in years to come.

## Methods

The goal of this analysis is the creation of high-resolution maps of ITN access, use, and NPC. Typically, such maps are generated via geospatial regression techniques alone, whereas this analysis first uses a mechanistic model to calculate mean national-level trends of access and NPC before regressing on access deviation, use gap, and NPC deviation (Fig. 1). The extra step of mechanistic modeling is necessary for two reasons. First, to incorporate non-survey data on net delivery and distribution that are crucial to constructing an accurate time series but would be challenging to incorporate into the traditional geospatial framework where time is handled via covariance structures. Second, because the primary outcome variables are often discontinuous in space or time, while regression techniques assume a smoothly varying output metric. By detrending the data, we were able to generate outcome variables with the appropriate mathematical properties.

### Definitions

*Net types*. Pre-treated nets with an effective insecticidal lifespan of at least 3 years are defined as "LLINs". Treated nets obtained or retreated within the past 12 months are defined as conventional ITNs, or "cITNs". LLINs and cITNs collectively comprise "ITNs", or simply "nets", as this analysis does not consider untreated nets.

*Net movement*. ITN "delivery" refers to manufacturer shipment of nets to national programs or other distributing bodies, while ITN "distribution" refers to the provision of nets to end users. Most countries have continuous ITN distribution channels through antenatal clinics and child immunization programs, which are supplemented every 3–4 years with mass distribution campaigns directly to households. Some countries distribute ITNs through schools or community-level systems in addition to or in place of mass campaigns. ITN "stock" refers to the number of nets available to distribute at a given time. Because programs may not immediately distribute all of the nets delivered to them, stock in a given year is not necessarily equal to reported manufacturer delivery counts.

ITN "crop" refers to the total number of nets in homes at a given time point. Crop depends upon both ITN distribution and "retention", the length of time for which nets are owned before being discarded or repurposed. Most countries plan mass net distributions every 3 years, under the assumption that average retention times are not much shorter than this[6,56]. "NPC" is ITN crop divided by population at risk.

*Net coverage*. A person is defined to have "access" to an ITN if they live in a household where they could sleep under an ITN, assuming two people per net[5]. Population-level access is the proportion of people with access. To avoid underestimating this metric, access is calculated at the individual level—i.e., in a household of ten people and three nets, six people will be defined as "having access" even though the household as a unit does not have sufficient nets for all its inhabitants. This is in contrast to "household-level access", which captures the proportion of households that own at least one ITN for every two people[5]. Access and NPC are calculated from household net counts and household sizes in nationally representative surveys. ITN "use" is the proportion of people who sleep under a net, measured in surveys by listing which people in a household slept under each net the night prior to the survey. For all survey-based metrics, the population denominator is calculated from "de facto household size", defined as the number of people who slept in the household the night prior to the survey. The "use rate", calculated as use divided by access, is the proportion of people with access to a net who slept under it. When calculated at the population level, all of these metrics use population at risk as a denominator, defined as anyone living in a zone with stable *Plasmodium falciparum* malaria transmission[7].

"Access deviation" is the difference between access in a specific location within a country and the national mean access. Access deviation is positive when local access is greater than the national mean, and negative when local access is below the national mean. The "use gap" is the difference between access and use in a given location. The use gap is positive when not everyone with access to a net sleeps under it, negative when more than two people sleep under a single net, and zero when everyone who has access to a net sleeps under it respecting the "two people per net" guideline. Similar to access deviation, NPC deviation is the difference between NPC in a specific location within a country and the national mean NPC.

**Stock and flow**. The "stock-and-flow" model is a discrete compartmental mechanistic model with a quarterly time step (Fig. 1, top). This model generates a net crop time series from three data sources: annual LLIN delivery data, annual distribution data, and sparse net crop data from nationally representative surveys. The main parameters estimated are quarterly ITN crop and median ITN lifespan for each country. For full methodological details, see Supplementary Note 1.4.

*Data*. National, annual LLIN delivery data from 2000 to 2019 were compiled by the Alliance for Malaria Prevention's Net Mapping Project (https://netmappingproject.allianceformalariaprevention.com/). No such data are available for cITNs. For cITNs over the full time period and for LLINs in 2020, net delivery was assumed to equal net distribution.

Complete national time series of cITN and LLIN distribution by year from 2000 to 2020 were compiled from three partially complete sources: net distribution reports from NMCPs to WHO (personal communication), data collected by the African Leaders Malaria Alliance (https://alma2030.org/), and prospective malaria operational plans from countries receiving PMI funding (https://www.pmi.gov/resource-library/mops/fy-2020). For complete details, see Supplementary Note 1.3.

Net crop data were extracted from 161 nationally representative household surveys conducted in sub-Saharan Africa since the year 2000. Ninety-five of these surveys included geolocated microdata at the household level and were additionally used to fit the spatiotemporal regression model (Supplementary Fig. 1.2 and Supplementary Table 1.1). All surveys include data on both cITNs and LLINs, except for some surveys conducted in the past 3 years in which all nets are assumed to be LLINs.

*Model details*. To estimate net crop at a given time point, the model enforces two rules:

(1) The number of nets distributed must be bounded by the available stock (upper) and the reported distribution counts (lower).
(2) Net loss must follow an S-shaped "smooth compact" curve as described in refs. [12,57]. This functional form was developed specifically to track net retention and is widely used in net durability studies[24,25,27–30,41,42,58]. A separate median retention time for this curve is estimated for every country based on the available survey data.

National net crop is converted to access via a household-size-based regression analysis. The stock-and-flow model was written in JAGS and run in R version 3.6.3 using the rjags package version 4.3.0. For full methodological details, see Supplementary Note 1.4.

**Spatiotemporal regression**. Having determined mean national time trends, geolocated values of access deviation, use gap, and NPC deviation were calculated

from survey data and used to fit a series of spatiotemporal regressions (Fig. 1, bottom). These deviation metrics are smoother than the full metrics and fulfill the stationarity requirements of this statistical technique (Supplementary Fig. 2.1). For full details of the regression model see Supplementary Note 2.3.

*Data*. Household-level data on net metrics were extracted from the 95 geolocated surveys as described in "Stock and flow" above, Supplementary Note 1.3.3, and Supplementary Fig. 1.2. Data were aggregated to the 5-by-5-km pixel level for analysis. The final dataset contained 34,352 data points covering 28 countries and 17 years.

Stochastic Partial Differential Equation (SPDE) regression models are widely used tools in geostatistical modeling[59]. Environmental and socioeconomic covariates at monthly, annual, and static temporal resolution were included to inform model fit (Supplementary Table 2.1). For more details on covariate selection, see Supplementary Note 2.5.

*Regression model*. For all three outcome metrics, Gaussian SPDE models were fitted in R using the R-INLA package version 20.03.17 (https://www.r-inla.org/). Access deviation and use gap were transformed via the empirical logit function to expand their domain from [−1, 1] to [−∞, ∞], and all outcome variables were transformed via the inverse hyperbolic sine function to facilitate fitting using a Gaussian likelihood. In addition to the fixed effects described in Supplementary Table 2.1, all models included a spatial random field with a Matérn covariance function and a first order autoregressive time component. After model fitting and prediction of access deviation and use gap, ITN access was calculated as the sum of national access and access deviation, ITN use was calculated as the sum of access and use gap, and NPC was calculated as the sum of national NPC and NPC deviation.

*Relative gain*. On the question of barriers to reaching coverage targets, there is ongoing debate in the malaria policy community about the relative importance of lacking access to a net versus not using a net to which one has access. This question was addressed by determining how much improvement in ITN use could be gained by maximizing ITN access versus maximizing the use rate (Supplementary Note 2.7).

**Reporting summary**. Further information on research design is available in the Nature Research Reporting Summary linked to this article.

## Data availability

**Input data**: The household-level survey data used in this analysis is publicly available from the DHS (https://dhsprogram.com/) and MICS (https://mics.unicef.org/) websites. The national-level-aggregated survey data were gleaned from reports available at the MIS website (https://www.malariasurveys.org/, see Supplementary Table 3.1 for links to specific reports). Data on manufacturer delivery of nets are available from the AMP Net Mapping Project (https://allianceformalariaprevention.com/working-groups/net-mapping/). Data on NMCP distribution of nets will be available via WHO and from ALMA in the coming months. Prospective distribution estimates from PMI reports are available at https://www.pmi.gov/resource-library/mops/fy-2020. All covariate data are available from the sources listed in Supplementary Table 2.1, and the specific versions used in this analysis can be found at https://malariaatlas.org/research-project/metrics-of-insecticide-treated-nets-distribution. **Results**: All results, including annual rasters of ITN access, use, use rate, and nets-per-capita with upper and lower bounds, are available at https://malariaatlas.org/research-project/metrics-of-insecticide-treated-nets-distribution. This page also contains an interactive uncertainty visualization, access to the covariate data used in this analysis, and other helpful links. Data used in the figures of this publication can be found, along with plotting code, at the GitHub repository in the "Code availability."

## Code availability

**Data collection**: Data from DHS were downloaded from https://dhsprogram.com/ using the code in https://github.com/harry-gibson/DHS-To-Database, and the ITN-relevant variables selected using the SQL queries available at https://github.com/harry-gibson/DHS-Data-Extractions/tree/main/ITN_Access_and_Use/SQL. Data from MICS surveys, MIS reports, net manufacturer deliveries, and NMCP distributions of nets were collected manually and cleaned using the code described in the "Data analysis." No additional software was used for data collection. **Data analysis**: All code used for data analysis and modeling is publicly available in the `publication-2021` branch of https://github.com/bertozzivill/map-itn-cube/tree/publication-2021, and in the corresponding release labeled `nat-comms-submission`. This repository also contains the data and code used to generate the figures in this paper. For analyses run on Google Cloud, Dockerfiles can be shared upon request. This analysis was run using R version 3.6.3, INLA version 20.03.17, rjags version 4.3.0, and rstan version 2.19.3.

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

## Acknowledgements

We extend our deepest gratitude first and foremost to the data collectors and coordinators of the household-level surveys upon which this analysis relies. We also thank national country programs for sharing their net distribution data, Melanie Renshaw and the African Leaders Malaria Alliance for supplementary distribution data, and John Milliner and the Alliance for Malaria Prevention's Net Mapping Project for net manufacturer data. We thank Thomas Smith and his team for kindly sharing their manuscript and insights on net durability and retention. We thank Abdisalan Noor and Beatriz Galatas of WHO for collating and delivering this programmatic data, as well as for their extensive insights, suggestions, and patience.

This publication is based on research conducted by the Malaria Atlas Project and funded in whole or in part by the Bill & Melinda Gates Foundation, including models and data analysis performed by the Institute for Disease Modeling at the Bill & Melinda Gates Foundation.

This work was supported, in whole or in part, by the Bill & Melinda Gates Foundation (OPP1197730). Under the grant conditions of the Foundation, a Creative Commons Attribution 4.0 Generic License has already been assigned to the Author Accepted Manuscript version that might arise from this submission. This work was also supported by the Telethon Trust, Western Australia.

## Author contributions

A.B.-V., S.B., and P.W.G. conceived the study. A.B.-V. and S.B. designed the models. C.A.B., H.K., D.J.W., K.E.B., S.F.R., and T.D.H. contributed domain knowledge and methodological improvements. C.V.-R., A.K.N., H.S.G., and J.H. processed the data and developed computational infrastructure. A.B.-V. wrote the first draft of the manuscript and created all visualizations. A.B.-V., C.A.B., H.K., D.J.W., C.V.-R., A.K.N., H.S.G., J.H., K.E.B., S.F.R., S.K., P.A., R.A., E.C., E.G.C., E.L.C., J.M., S.M., J.R., T.S., K.A.T., T.D.H., P.W.G., and S.B. interpreted the results, contributed to writing, and approved the final version for submission.

## Competing interests

The authors declare no competing interests.
