## [Peer Review File · Nature Communications]

REVIEWERS' COMMENTS

Reviewer #1 (Remarks to the Author):

This work presents a comprehensive description of insecticide-treated net (ITN) availability and use in sub-Saharan Africa for 2000-2020. The work incorporates a variety of data sources and carefully integrates the different data types and quantity. Importantly, the extensive methods section (in the appendix) is thorough and easy to follow. In addition, the code is publicly available. While in all modeling work many choices can be debated, this work does a clear and extensive job of explaining the reasoning behind their choices, providing uncertainty analysis, and stating how their assumptions likely impact results. Finally, this work addresses an important topic and has the potential to assist in policy decisions.

The manuscript would be improved with clearer definitions throughout (see minor comments). For example, what is meant by terms such as "access" and "at risk".

Minor comments:

- P2, L16-17 The clause "have flowed into ITN production and distribution" goes with "two billion nets" as written but more make more sense associated to "representing billions of dollars of aid" This sentence should be revised accordingly.
- P2, L22: What is meant by 'supplements' data?
- P2, L31: How is "at risk" defined here?
- P2, L37: Does the 80% modify access or use or both?
- P3, L50-51: Reference to justify that these alternative uses are not at "sufficient scale to cause concern"?
- P3, L79: Missing space in "percapita"
- P5, L92-93: Are these percentages continental or averaged across countries?
- P5, L97: State why 80% is chosen as the comparison.
- P5, L97: Add "Only" at the beginning of the sentence "Fourteen ..."
- P7, L129: Define DHS, MIS
- P7 (and onward): Clearly indicate with figures are in the supplement.
- P11, Figure 5: Difficult to read which bar is associated with which country.
- P14, L242-244: Where is demonstration of the association of lower use rates and time/geographies of lower actual or perceived risk?
- P16, L317-319: This is the first definition of "at-risk" Move this into the introduction.
- P17, L357: Mean national-level trends of what?
- P18, L390-391: This definition of access involving two people per net should come earlier.
- P19, L418: Where can more information on this model be found? Is it discrete?
- P20, L456: Talk more about the S-shaped curve in this manuscript.
- P21, L459: Clarify which surveys are being referred to and in which section.

Reviewer #2 (Remarks to the Author):

Bertozzi-Villa et al. have generated high resolution spatial-temporal maps for evaluating insecticide treated net (ITN) coverage in Sub-Saharan Africa based on ITN access, use, and nets-per-capita. Through the combination of a stock-and-flow model and a geospatial model, the authors compared estimates of different aspects of ITN coverage over time as well as across and within countries. Their model is able to capture subnational heterogeneity in net coverage due to distribution strategies and/or accessibility challenges as well as seasonal changes in net use. They point out that net retention time in most countries is actually shorter than that assumed by the WHO, which could have significant ramifications for malaria control strategies. Finally, their model indicates that the largest challenge to universal coverage is increasing global access than ITNs, as opposed to increasing ITN use. This work fills a gap in the literature by providing a standardized approach to estimating the history of net coverage over the past 20 years across 40 countries and will continue to serve as an excellent framework moving forward. Furthermore, these maps will be a resource of high interest to those working to improve the control of/eliminate malaria. The

methods and data are very well documented, both in the supplement and online, and their assumptions and limitations are clearly stated. I would recommend this manuscript for publication.

Reviewer #3 (Remarks to the Author):

The manuscript addresses an important data gap on the access and use of insecticide-treated nets (ITNs) in Africa. Overall, this is a well-conducted research with methods being clearly described, results being carefully discussed, and limitations being explicitly explained. I have little to add but would suggest the authors to extend their abstract and discussion to include explanations on the driving factors underpinning their three main findings.

Reviewer 1

Comment

This work presents a comprehensive description of insecticide-treated net (ITN) availability and use in sub-Saharan Africa for 2000-2020. The work incorporates a variety of data sources and carefully integrates the different data types and quantity. Importantly, the extensive methods section (in the appendix) is thorough and easy to follow. In addition, the code is publicly available. While in all modeling work many choices can be debated, this work does a clear and extensive job of explaining the reasoning behind their choices, providing uncertainty analysis, and stating how their assumptions likely impact results. Finally, this work addresses an important topic and has the potential to assist in policy decisions.

The manuscript would be improved with clearer definitions throughout (see minor comments). For example, what is meant by terms such as “access” and “at risk”.

[Minor comments logged in table below.]

Response

We thank the reviewer for their thoughtful comments and suggestions. Please see notes and responses in the table below.

No.	Comment	Response
1	P2, L16-17 The clause “have flowed into ITN production and distribution” goes with “two billion nets’ as written but more make more sense associated to “representing billions of dollars of aid” This sentence should be revised accordingly.	Completed, thank you for the suggestion.
2	P2, L22: What is meant by ‘supplements’ data?	“supplements” changed to “fills gaps” for clarity.
3	P2, L31: How is “at risk” defined here?	Clarifying language added, thank you.
4	P2, L37: Does the 80% modify access or use or both?	“for both” added for clarity.
5	P3, L50-51: Reference to justify that these alternative uses are not at “sufficient scale to cause concern”?	“a small proportion of” added for clarity, the references cited earlier in the sentence provide justification for this statement.
6	P3, L79: Missing space in “percapita”	Dash added, thank you.
7	P5, L92-93: Are these percentages continental or averaged across countries?	All metrics in this paragraph (except absolute net

		counts) are population-weighted averages. Language was changes from “continental” to “aggregating across all countries in this analysis” for clarity.
8	P5, L97: State why 80% is chosen as the comparison.	Done, thank you.
9	P5, L97: Add “Only” at the beginning of the sentence “Fourteen ...”	We chose not to use the term “only” in this statement as it could indicate a judgmental stance toward country-level performance, which we are not in a position to comment on.
10	P7, L129: Define DHS, MIS	Done, thank you.
11	P7 (and onward): Clearly indicate with figures are in the supplement.	All references to the supplement have been stated more explicitly, thank you.
12	P11, Figure 5: Difficult to read which bar is associated with which country.	We have increased contrast on the country labels to improve readability. This figure’s data is also replicated in table form (Supplementary Table A.5) for convenience. The legend has been updated to indicate this.
13	P14, L242-244: Where is demonstration of the association of lower use rates and time/geographies of lower actual or perceived risk?	The literature referenced in this sentence and the previous one provides this demonstration, as the comparison between risk and net use is not in the scope of this analysis.
14	P16, L317-319: This is the first definition of “at- risk” Move this into the introduction.	Clarifying language added to paragraph 2 of the introduction, thank you.
15	P17, L357: Mean national- level trends of what?	“of access and NPC” added for clarity, thank you.
16	P18, L390-391: This definition of access involving two people per net should come earlier.	“Access” is first defined in the introduction (Second paragraph, first sentence).
17	P19, L418: Where can more information on this model be found? Is it discrete?	“discrete” and supplementary section numbers added for clarity.
18	P20, L456: Talk more about the S -shaped curve in this manuscript.	Additional information and citations added, thank you.
19	P21, L459: Clarify which surveys are being referred to and in which section.	Section references added, thank you.

Reviewer 2

Comment

Bertozzi-Villa et al. have generated high resolution spatial-temporal maps for evaluating insecticide treated net (ITN) coverage in Sub-Saharan Africa based on ITN access, use, and nets-per-capita. Through the combination of a stock-and-flow model and a geospatial model, the authors compared estimates of different aspects of ITN coverage over time as well as across and within countries. Their model is able to capture subnational heterogeneity in net coverage due to distribution strategies and/or accessibility challenges as well as seasonal changes in net use. They point out that net retention time in most countries is actually shorter than that assumed by the WHO, which could have significant ramifications for malaria control strategies. Finally, their model indicates that the largest challenge to universal coverage is increasing global access than ITNs, as opposed to increasing ITN use. This work fills a gap in the literature by providing a standardized approach to estimating the history of net coverage over the past 20 years across 40 countries and will continue to serve as an excellent framework moving forward. Furthermore, these maps will be a resource of high interest to those working to improve the control of/eliminate malaria. The methods and data are very well documented, both in the supplement and online, and their assumptions and limitations are clearly stated. I would recommend this manuscript for publication.

Response

We thank the reviewer for their time and endorsement of this work.

Reviewer 3

Comment

The manuscript addresses an important data gap on the access and use of insecticide-treated nets (ITNs) in Africa. Overall, this is a well-conducted research with methods being clearly described, results being carefully discussed, and limitations being explicitly explained. I have little to add but would suggest the authors to extend their abstract and discussion to include explanations on the driving factors underpinning their three main findings.

Response

We thank the reviewer for this suggestion. We have expanded the abstract to include driving factors, added comments on paragraph 3 of the discussion (lines 225 onward) to give more detail on net misallocation, and added comments on paragraph 5 of the discussion (lines 244 onward) with additional reasons for a high use rate.